# HarmonicNeRF: Geometry-Informed Synthetic View Augmentation for 3D Scene Reconstruction in Driving Scenarios

Xiaochao Pan*
Department of Software, Taiyuan
University of Technology
Taiyuan, China
panxiaochao1369@link.tyut.edu.cn

Jiawei Yao*
School of Engineering and
Technology, University of
Washington
Tacoma, United States
jwyao@uw.edu

Hongrui Kou*
Department of Vehicle Engineering,
Jilin University
Changchun, China
kouhr23@mails.jlu.edu.cn

Tong Wu
School of Engineering and
Technology, University of
Washington
Tacoma, United States
tw96@uw.edu

Canran Xiao†
School of Business, Central South
University
Changsha, China
xiaocanran@csu.edu.cn

## Abstract

In the realm of autonomous driving, achieving precise 3D reconstruction of the driving environment is critical for ensuring safety and effective navigation. Neural Radiance Fields (NeRF) have shown promise in creating highly detailed and accurate models of complex environments. However, the application of NeRF in autonomous driving scenarios encounters several challenges, primarily due to the sparsity of viewpoints inherent in camera trajectories and the constraints on data collection in unbounded outdoor scenes, which typically occur along predetermined paths. This limitation not only reduces the available scene information but also poses significant challenges for NeRF training, as the sparse and path-distributed observational data leads to under-representation of the scene's geometry. In this paper, we introduce HarmonicNeRF, a novel approach for outdoor self-supervised monocular scene reconstruction. HarmonicNeRF capitalizes on the strengths of NeRF and enhances surface reconstruction accuracy by augmenting the input space with geometry-informed synthetic views. This is achieved through the application of spherical harmonics to generate novel radiance values, taking into careful consideration the color observations from the limited available real-world views. Additionally, our method incorporates proxy geometry to effectively manage occlusion, generating radiance pseudo-labels that circumvent the limitations of traditional image-warping techniques, which often fail in sparse data conditions typical of autonomous driving environments. Extensive experiments conducted on the KITTI, Argoverse, and NuScenes datasets demonstrate our approach establishes new benchmarks in synthesizing novel depth views and reconstructing

scenes, significantly outperforming existing methods. Project page: https://github.com/Jiawei-Yao0812/HarmonicNeRF

## CCS Concepts

• **Computing methodologies → Computer vision**; **Computer graphics**.

## Keywords

Neural Radiance Fields, Sparse Views, Ray Augmentation, Autonomous Driving

### ACM Reference Format:

Xiaochao Pan, Jiawei Yao, Hongrui Kou, Tong Wu, and Canran Xiao. 2024. HarmonicNeRF: Geometry-Informed Synthetic View Augmentation for 3D Scene Reconstruction in Driving Scenarios. In *Proceedings of the 32nd ACM International Conference on Multimedia (MM '24), October 28-November 1, 2024, Melbourne, VIC, Australia.* ACM, New York, NY, USA, 10 pages. https://doi.org/10.1145/3664647.3680851

## 1 Introduction

In recent years, with the rapid advancement of autonomous driving technology, 3D reconstruction has become a crucial component for ensuring precision in navigation and overall safety. Amidst this technological evolution, Neural Radiance Fields (NeRF) [30] have emerged as a groundbreaking 3D reconstruction technique, gaining widespread attention for their ability to generate highly detailed and accurate models of complex environments. Unlike traditional Multi-View Stereo (MVS) [7, 14, 52] methods, NeRF utilizes deep learning [9, 10] to model environments, capable of synthesizing continuous and photorealistic images, thereby significantly enhancing reconstruction quality and visual fidelity. Furthermore, NeRF's capability to synthesize intricate autonomous driving scenes [18, 23] enriches training datasets for autonomous systems, aiding in the improvement of their generalizability and decision-making quality.

Despite the remarkable performance of NeRF and its derivatives in targeted synthetic rendering datasets, their scene reconstruction performance significantly diminishes when applied to specific autonomous driving datasets such as KITTI [16] and NuScenes [4].

---

*These authors contribute equally to this work.
†Corresponding author.

MM '24, October 28-November 1, 2024, Melbourne, VIC, Australia

ACM ISBN 979-8-4007-0686-8/24/10
https://doi.org/10.1145/3664647.3680851

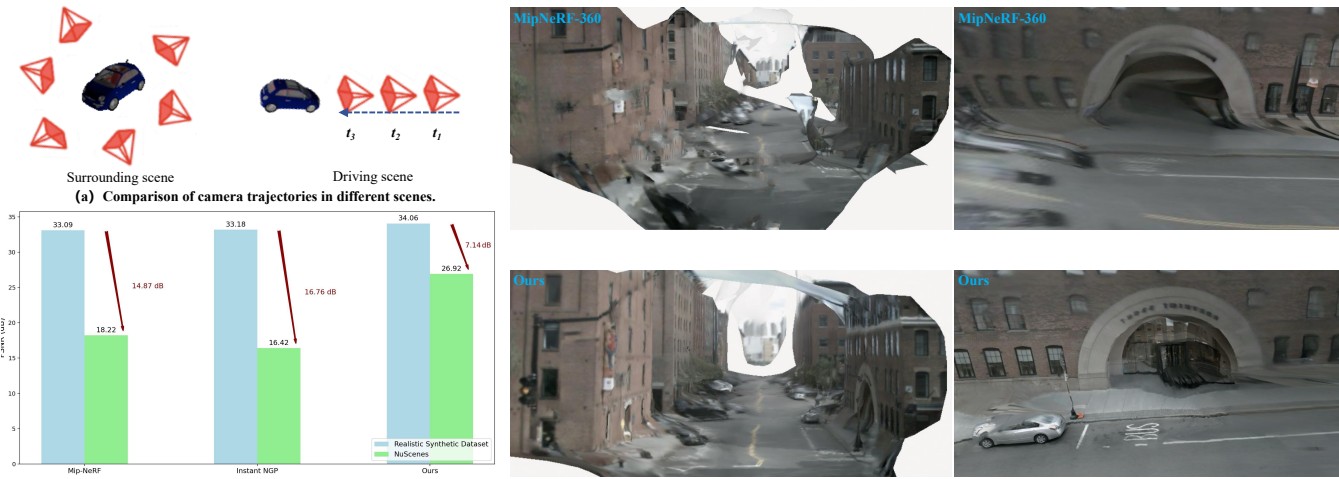

Figure 1: (a) illustrates the differing camera paths in both general surrounding scenes and specific driving scenarios. The camera trajectories in driving scenes are depicted as more linear and path-constrained, indicative of the typical movement patterns in autonomous driving data collection, as opposed to the more varied viewpoints found in general scenes. (b) shows our method's effectiveness in dealing with the challenges of sparse and dynamic driving environments. Our reconstruction in (c) demonstrates significantly clearer and more accurate geometries

This decline is primarily attributed to the limited viewpoints provided by autonomous driving scenarios, failing to offer sufficient perspective information for high-quality synthesis. Additionally, the camera trajectories in autonomous driving are characterized by sparse viewpoints of single self-motion, differing from the object-centered forms previously encountered. The presence of moving objects in autonomous driving scenes introduces variations in lighting and geometry over time, resulting in artifacts in synthesized images and a decrease in reconstruction effectiveness. The original NeRF model does not account for moving objects, limiting its application in autonomous driving contexts.

To address the limitations of NeRF in modeling dynamic scenes undergoing motion, several scholars have proposed extensions to the NeRF network architecture. Innovations such as D-NeRF [37] and NRNeRF [43] for scenes with non-rigid deformations, and time-variable dynamic radiance fields like NSFF [24] , NeRFlow [12], DynamicNeRF [15], and RoDynRF [25] have been introduced. These adaptations incorporate temporal dimensions or consider non-rigid transformations to accommodate scene dynamics. Moreover, concerns regarding the original NeRF model's size and its direct application to autonomous driving scenarios leading to significant artifacts and reduced visual fidelity have led to the development of solutions like Mip-NeRF [1], which considers positions as conical sections of a light ray rather than points to mitigate aliasing effects on NeRF performance. NeRF in the Wild [27] addresses varying environmental conditions by encoding appearances.

Although these methods have somewhat enhanced NeRF's capability in dynamic scene reconstruction, they generally fail to fully address the challenges posed by the sparse viewpoints and path-distributed data collection inherent in autonomous driving scenes. In response to these issues, this paper introduces a novel geometry-guided ray augmentation technique specifically designed for sparse view scene reconstruction in driving scenarios. Our approach not only effectively manages challenges presented by dynamic objects

and sparse viewpoints but also significantly improves scene reconstruction accuracy from sparse views through the innovative application of proxy geometry and spherical harmonics. Experiments conducted on challenging autonomous driving datasets such as KITTI, Argoverse [5], and NuScenes validate the superiority of our method, establishing new benchmarks in 3D scene reconstruction for the domain of autonomous driving. In summary, our contributions are as follows:

- We leverage spherical harmonics to comprehensively integrate all color observations at a 3D point. This approach enables the generation of pseudo-labels that align with the natural distribution of radiance, enhancing the consistency and accuracy of scene illumination and texture.
- We introduce a novel use of proxy geometry to address occlusion challenges during the ray augmentation process. This strategy ensures that point radiance is unconfounded, facilitating the reconstruction of more accurate surfaces from sparse viewpoints, a common scenario in autonomous driving environments.
- Our method is designed as a versatile, plug-and-play solution, compatible with existing sparse implicit neural surface reconstruction techniques. It demonstrates exceptional performance in sparse view reconstruction tasks, achieving superior results in driving scenarios without requiring additional data or extensive pre-training.

## 2 Related Work

**Neural Surface Reconstruction.** Recently, neural implicit representations have demonstrated superior effectiveness in various tasks such as 3D object [8, 28, 36] and scene representation [30, 39, 40], novel view synthesis [2, 30, 47] and multi-view 3D reconstruction [33, 34, 48, 54]. Among them, NeRF and its variants [2, 30] demonstrated state-of-the-art performance for novel view synthesis

with differentiable volumetric rendering. However, they require dense images for input and it is hard to extract high-quality geometry because the predicted density field lacks sufficient constraints. Other works utilizing surface rendering [33] produce more accurate modeling but require additional constraints such as ground truth masks or depth priors for supervision [54, 56]. Several later studies combine the advantages of surface and volume representation, generating satisfying surfaces with only image inputs [34, 48, 53]. For example, NeuS [48] combined volume rendering with signed distance functions to circumvent the problem of insufficient surface constraint in the original NeRF. The sampling efficiency was further improved to allow for reconstructing in the wild scenes [41]. UNISURF [34] also unifies surface and volume rendering but with occupancy fields. Similar to them, our work also only requires multi-view images for 3D reconstruction.

**Large-Scale Scene Reconstruction.** NeRF's static assumptions falter in the dynamic, expansive scenes of autonomous driving, where direct application incurs artifacts and fidelity loss due to lighting and occlusion changes. Tackling NeRF's capacity limitations, methods like Mega-NeRF [45] and Block-NeRF [42] divide scenes into smaller, manageable NeRF segments, improving training efficiency and computational feasibility. Mip-NeRF [1] addresses aliasing by representing positions as conical sections rather than points, enhancing rendering quality. Approaches like Block-NeRF's adaptation of NeRF in the Wild [27] encode varying appearances for realistic lighting condition reconstructions. Recent extensions such as LocalRF [29], READ [23], and S-NeRF [50] further NeRF's applicability in autonomous driving, dealing with changing geometries and appearances characteristic of these environments.

**Implicit Representation with Sparse Views.** In the realm of autonomous driving, reconstructing environments from sparse views is a pressing challenge due to the sporadic nature of data acquisition on the road. Two primary strategies have emerged to adapt NeRF for sparse input: leveraging pre-trained CNNs to extract image features from multi-view inputs or using sparse point clouds from Structure-from-Motion (SfM) for additional supervision.

PixelNeRF [55] innovatively simulates continuous neural scenes from a few input images, employing pre-trained layers of convolutional neural networks (CNNs) and bilinear interpolation. This method enhances the feature extraction process for each sampling point, fully utilizing the characteristics of input images. It then conveys the extracted feature points, spatial locations, and viewing directions to the NeRF network, enabling the construction of continuous static scenes from sparse image sets [55]. The General Radiance Field (GRF) [44] adopts a similar approach to PixelNeRF but differs in that it operates in a standardized canonical space, offering more generality and versatility, particularly under the varying conditions encountered in autonomous driving scenarios. Point-NeRF [51] combines the explicit representation of point clouds with the implicit NeRF technique. It harnesses the strengths of both 3D representation forms and adapts efficiently to the surfaces of scenes, a crucial feature for capturing the complexities of driving environments [6].

Other methods such as GeoNeRF [20] and DietNeRF [19] propose novel supervision techniques that enrich the input data for NeRF

with semantic consistency or spatial geometry from unobserved viewpoints. These methods mitigate issues like floating artifacts that are common when dealing with sparse views but often require additional datasets for training, which may introduce domain generalization challenges. HarmonicNeRF bypasses the need for pre-training entirely, diverging from the aforementioned works. Instead of regularization or relying on scene priors, we focus on fitting the radiance distribution for each surface point, using it as augmented information. We posit that the radiance distribution at a point is a harmonic function, decomposable into a spherical harmonic expansion. This insight leads us to develop a physically grounded pipeline to generate precise pseudo labels for radiance supervision, tailor-made for the dynamic and variable conditions present in autonomous driving data collection [11, 22].

## 3 Preliminaries: NeuS

NeuS [48] represents a 3D scene as two continuous functions both parametrized by a multi-layer perception (MLP). The first one is a signed-distance function that takes as input 3D position $\mathbf{x} = (x, y, z)$ and predicts its signed distance: $f_1(\mathbf{x}) \rightarrow s$, and the other takes both $\mathbf{x}$ and its viewing direction $\mathbf{d}$ and predicts its view-dependent color: $f_2(\mathbf{x}, \mathbf{d}) \rightarrow \mathbf{c}$.

Similar to NeRF [30], the color of a pixel in NeuS [48] only depends on the radiance along a ray with no other lighting factors. For a camera ray $\mathbf{r}(t) = \mathbf{o} + t\mathbf{d}$ with center $\mathbf{o}$ and direction $\mathbf{d}$, its color can be derived with volume rendering [21]. In practice, NeuS samples $N$ points along $\mathbf{r}(t)$ with $t = t_1, ... t_N, t_i < t_{i+1}$ and approximate the per-pixel color with the following:

$$\hat{\mathbf{C}}(\mathbf{r}) = \sum_{i=1}^{N} T_i \alpha_i \mathbf{c}_i, \qquad (1)$$

$$T_i = \prod_{j=1}^{i-1} (1 - \alpha_j), \qquad (2)$$

where $\alpha_i$ is the discrete opacity value that is a function of the probability density function of $f_1$, and $T_i$ is the accumulated transmittance which indicates the probability that a ray traverses from $t_1$ to $t_i$ without hitting any particle. NeuS exploits volume rendering to minimize the mean-squared error (MSE) between the predicted ray color $\hat{\mathbf{C}}(\mathbf{r})$ and the ground truth $\mathbf{C}(\mathbf{r})$:

$$\mathcal{L}_{\text{MSE}} = \sum_{\mathbf{r}} \|\hat{\mathbf{C}}(\mathbf{r}) - \mathbf{C}(\mathbf{r})\|_2^2. \qquad (3)$$

Besides color loss, it also leverages an L1 regularization and a mask loss if available. With successful minimization, the formulation of NeuS enables the network to learn a signed-distance function that represents an accurately reconstructed surface.

## 4 Methods

When limited images and corresponding training rays are given, from Section 3, only a small proportion of the 3D space has supervision, while the results of the rest space purely rely on the interpolation ability of the network. Therefore, the main goal of our paper is to augment the supervision signal of Implicit Neural Representation (INR) by inferring rays from unseen viewpoints based on limited inputs and the reconstructed INR in an accurate

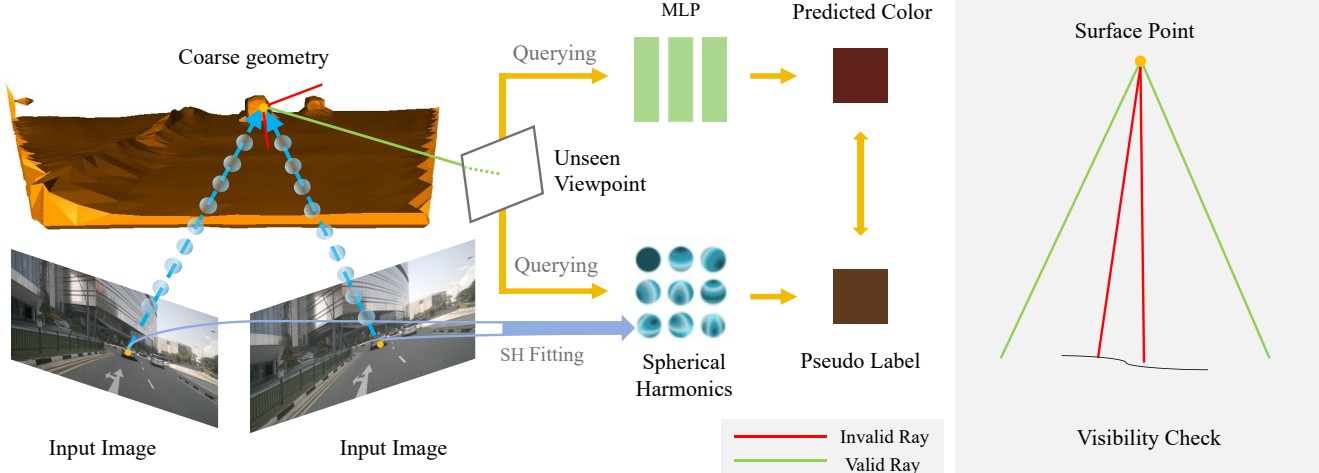

**Figure 2: Method Overview. We exploit the coarse geometry in radiance field training to guide its augmentation with sparse inputs. (left) For a surface point v, we aggregate the color observations from all inputs to fit a spherical harmonics expansion, thus the pseudo labels for all the augmented rays passing through v can be obtained through querying the SH, (right) when generating augmented rays, we check their visibility and exclude those cannot be actually observed.**

way. Particularly, we found that model the problem of inferring unseen radiance as fitting the radiance maps for spatial points, and the radiance map of each point could be estimated from the known radiance of that point based on our prior radiance map function.

In this work, we introduce HarmonicNeRF: Enhancing Neural Radiance Fields with Visibility-Driven Spherical Harmonics, a novel approach that significantly advances the capabilities of neural radiance fields by incorporating spherical harmonics for accurate radiance map estimation and employing visibility checks alongside surface ray casting for improved scene reconstruction from sparse viewpoints.

In the following of this section, we introduce the details of our method. Our radiance augmentation algorithm is composed of three steps: Surface Ray Casting in Section 4.1, Visibility Checking in Section 4.2, and Radiance Map Estimation in Section 4.3.

### 4.1 Surface Ray Casting

Same to NeRF, the input to our method is a set of posed images $\{I_i\}_{i=1}^{N}$, where $N$ is the number of images. The original training set is denoted by $\mathcal{R} = \{(\mathbf{r}, \mathbf{c})\}$, in which $\mathbf{r}$ and $\mathbf{c}$ are the ray directions and ray color. Since our method relies on the rough scene geometry, we first train a radiance field for a few thousand steps and construct the mesh with marching cubes [26]. The vertices of this rough mesh are regarded as points lying on the surface. Then starting from these surface points $\mathbf{v}$, we shoot a number of rays $\mathbf{r}_{rand} \in \mathcal{R}$ in random directions. We denote the ray casted as $\tilde{\mathcal{R}}_{\mathbf{v}}$. We ensure that the angle between new rays and the surface normal of $\mathbf{v}$ is less than $\frac{\pi}{2}$, so these candidates for augmented rays are confident.

### 4.2 Visibility Check

However, some rays in $\tilde{\mathcal{R}}_{\mathbf{v}}$ might not actually be observed from a valid viewport since they will hit the mesh and get occluded. Figure 2 shows an example of this case. To filter out those invalid rays, we include an additional visibility check step to exclude those invalid rays. Specifically, for each ray $\mathbf{r}_{aug}$, we march from the

starting point $\mathbf{v}$ according to the scene signed-distance field (SDF) and terminate when the surface is reached or achieve the maximum number of steps. Since the ray marching method is efficient, we integrate it directly into the training pipeline without any offline computation.

---

**Algorithm 1** Ray Marching for Visibility Check

**Input**: pos (starting point), $sdf$ (signed-distance function)
**Output**: visible or not

1: step := 0
2: d := $sdf$(pos)
3: **while** d > $\epsilon$ and step < max_steps **do**
4:      pos = pos + dir * d
5:      d = $sdf$(pos)
6:      step = step + 1
7: **end while**
8: **if** step < max_steps **then**
9:      return true  // visible
10: **else**
11:      return false  // invisible
12: **end if**

---

### 4.3 Radiance Map

After the visibility check, rays in $\tilde{\mathcal{R}}_{\mathbf{v}}$ are ensured to be observed from at least one camera viewpoint. The next question is what's the proper radiance value of rays in $\tilde{\mathcal{R}}_{\mathbf{v}}$? In previous ray augmentation methods [57], a surface ray has only one known radiance value, since the source view visibility check is not performed, thus they simply assign a color from one input view, which neglects the view-dependent effect and leads to inconsistent pseudo label. However, as shown in the previous discussion, a point in the scene surface could be seen from multiple views and we actually have more information regarding the radiance of surface point $\mathbf{v}$. Thus, a main contribution of our method illustrated as follows is that we utilize all reasonably credible information to reconstruct the radiance

distribution of surface point $\mathbf{v}$ to infer the radiance of novel rays. Actually, not only did we fit the radiance distribution but also we achieved extrapolation using spherical harmonics expansion fitting.

Our method handles the inference of radiance in two situations. Firstly in the simpler case, if the point can only be seen from less than $N_v$ views ($N_v$ is empirically 10 in our experiment), we think that we don't have enough information to reconstruct the radiance distribution. Thus, the radiance value of a randomly chosen view is assigned to all new rays origin from point $\mathbf{v}$. This is basically in accordance with the previous method.

If the point $\mathbf{v}$ under processing can be viewed from more than $N_v$ views, then in this case we try to infer a distribution of radiance over all viewing directions. Our solution to this problem is to fit a spherical harmonic expansion, which is widely used in precompute radiance transfer and environment maps in computer graphics. We represent the radiance distribution using spherical harmonics:

$$\mathbf{c_v}(\mathbf{d}) = \sum_{\ell=0}^{\ell_{\max}} \sum_{m=-\ell}^{\ell} k_\ell^m Y_\ell^m(\mathbf{d}), \tag{4}$$

where $\mathbf{c_v}(\mathbf{d})$ is the radiance of point $\mathbf{v}$ viewing from $\mathbf{d}$. Namely, for vertice $\mathbf{v}$, we back-project it to $\{I_i\}_{i=1}^N$ and filter views that according to visibility (use the same visibility check function as in 4.2). The color observations for $\mathbf{v}$ can thus be represented by:

$$\mathbf{S_v} = \{(\mathbf{c}_j, \mathbf{d}_j)\} = \{(\mathbf{c}_{v,j}, \text{Norm}(\mathbf{o}_j - \mathbf{v}))\}_{j=1}^{N_{\text{vis}}}, \tag{5}$$

where $N_{\text{vis}}$ is the number of views from where $\mathbf{v}$ can be seen and $\mathbf{o}_j$ is the corresponding camera center. These radiance sample points are used to fit a radiance map by computing the coefficients for the SH expansion via least squares fitting:

$$\arg\min_{k_\ell^m} \sum_{j=1}^{N_{vis}} |\mathbf{c_v}(\mathbf{d}_j) - \mathbf{c}_j|^2, \tag{6}$$

where $k_\ell^m \in \mathbb{R}^3$ is a set of 3 coefficients for RGB components. Then, for each ray in $\tilde{\mathcal{R}}_\mathbf{v}$, color is determined by querying the SH functions $Y_\ell^m$ given its viewing direction $\mathbf{d}$. Using such an SH expansion to fit the radiance distribution can provide extrapolation for novel view direction.

To summarize, our method extends the training rays of NeRF considering scene geometry. For each surface vertice $\mathbf{v}$, we first construct $\tilde{\mathcal{R}}_\mathbf{v}$ by shooting random ray directions from $\mathbf{v}$ and filter invisible ones through visibility check. Since $\mathbf{v}$ can be observed at multiple input viewpoints, we model it as an SH which thus can be queried at any augmented ray direction going through $\mathbf{v}$.

## 4.4 Depth Warping

To best utilize the available information in the known view, we propose to propagate the depth information to other views through image warping. For pixel $p_i(x_i, y_i)$ in reference view $I_{ref}$, the corresponding pixel $p_j(x_j, y_j)$ in the $j^{th}$ unseen view $I_{unseen}$ can be formulated as

$$p_j = K_{unseen} T \left( K_{ref}^{-1} Z_i p_i \right), \tag{7}$$

where $Z_i$ is the available depth of reference view, $T$ refers to the relationship between camera extrinsic matrices from $I_{ref}$ to $I_{unseen}$, and $K_{ref}$ and $K_{unseen}$ refer to the camera intrinsic matrices. We

further adopt the Painter's Algorithm when multiple points in the reference view are projected to the same point in the unseen view and select the point with the smallest depth as the warping result.

Through image warping, we can obtain a depth map of an unseen view, which can serve as a pseudo ground truth. Nevertheless, there is still an unavoidable gap between this pseudo ground truth and the real correspondence, since small misalignment in the predicted depth map can cause large errors when projected to other views. Moreover, it is quite common that the projected results contain some uncertain regions due to occlusion. To regularize the uncertain regions in the warped results, we utilize the self-supervised inverse depth smoothness loss, which uses the second-order gradients of the RGB pixel value to encourage the smoothness of the predicted depths:

$$\mathcal{L}_{smooth}(d_i) = e^{-\nabla^2 \mathcal{I}(\mathbf{x}_i)} \left( |\partial_{xx} d_i| + |\partial_{xy} d_i| + |\partial_{yy} d_i| \right), \tag{8}$$

where $d_i$ is the depth map, $\nabla^2 \mathcal{I}(\mathbf{x}_i)$ refers to the Laplacian of pixel value at location $x_i$.

## 5 Experiments

## 5.1 Novel Radiance Predicting from Spherical Harmonics

Firstly, we conduct an experiment to validate the effective method for generating radiance from unseen views. For predicting the novel radiance, we compare our method with a naive baseline: spherical linear interpolation, which interpolates the radiance of the unseen view from the nearest two seen views in the spherical coordinate system. Specifically, we implement the linear interpolation of radiance using the geodesic distance as the weight:

$$\hat{\mathbf{c}} = w\mathbf{c}_0 + (1 - w)\mathbf{c}_1, \tag{9}$$

$$w = \frac{d(\hat{\mathbf{v}}, \mathbf{v}_1)}{d(\mathbf{v}_1, \hat{\mathbf{v}}) + d(\hat{\mathbf{v}}, \mathbf{v}_2)}, \tag{10}$$

$$d(\mathbf{v}_1, \mathbf{v}_2) = \arctan \frac{|\mathbf{v}_1 \times \mathbf{v}_2|}{\mathbf{v}_1 \cdot \mathbf{v}_2}, \tag{11}$$

while $\hat{\mathbf{c}}, \hat{\mathbf{v}}$ are the radiance and view direction vector of novel view, $\{\mathbf{c}_i, \mathbf{v}_i | i \in 1, 2\}$ are these of the nearest two views, respectively. Intuitively, the spherical harmonics fitting method can provide extrapolation of radiance compared to the linear interpolation method while preserving the smoothing variation under the prior of harmonic function.

We generate a set of around $10,000$ points with all of their visible colors from the KITTI dataset [17] for evaluating the two methods for radiance prediction. The radiance from different views of each point is divided into two training (known) and testing (unknown) views with ratios from $8:2$ to $5:5$, in order to evaluate the performance under the different number of novel views. We report the average normalized Mean Square Error (MSE) in the novel views in Table 2. The average MSE for SH fitting is less than that of interpolation, especially when the number of novel views exceeds the number of known views. A visualization case is shown in Figure 3.

## 5.2 Implementation Details

**Dataset** In our experiments, we utilized three datasets known for their extensive capture of real-world driving scenarios: KITTI [17],

**Table 1: Quantitative comparison with selected methods on the KITTI dataset. The best and the second best results are shown in bold and underlined forms, respectively.**

| Methods | Dense | | | | Sparse | | | |
|---|---|---|---|---|---|---|---|---|
| | PSNR↑ | SSIM↑ | LPIPS↓ | ABSREL↓ | PSNR↑ | SSIM↑ | LPIPS↓ | ABSREL↓ |
| NeRF [30] | 20.03 | 0.652 | 0.502 | 0.203 | 15.07 | 0.552 | 0.603 | 0.253 |
| NSG [35] | 20.56 | 0.664 | 0.482 | 0.192 | 15.54 | 0.564 | 0.582 | 0.244 |
| pixelNeRF [55] | 19.48 | 0.631 | 0.518 | 0.213 | 14.56 | 0.534 | 0.614 | 0.261 |
| SUDS [46] | 20.14 | 0.643 | 0.493 | 0.198 | 15.12 | 0.547 | 0.598 | 0.248 |
| MARS [49] | 20.43 | 0.658 | 0.478 | 0.187 | 15.38 | 0.558 | 0.573 | 0.237 |
| Urban-NeRF [38] | 20.72 | 0.678 | 0.457 | 0.184 | 15.76 | 0.572 | 0.553 | 0.226 |
| MipNeRF-360 [3] | 21.99 | 0.692 | 0.437 | 0.088 | 16.93 | 0.589 | 0.498 | 0.144 |
| NeRF++ [58] | 20.29 | 0.520 | 0.585 | 3.917 | 17.60 | 0.535 | 0.562 | 4.960 |
| Instant-NGP [31] | 20.51 | 0.630 | 0.460 | **0.507** | 15.44 | 0.499 | 0.536 | 0.793 |
| Ours | **22.52** | **0.711** | **0.401** | 0.087 | **19.04** | **0.672** | **0.351** | **0.092** |

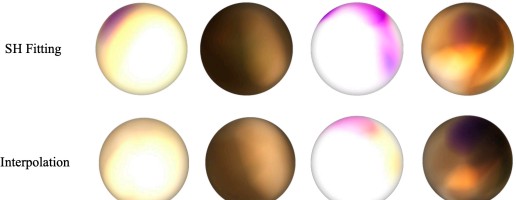

**Figure 3: Visualization of radiance fitting. We project the radiance value of points distributed on the upper hemisphere into this circle to get a visualization of the fitted radiance distribution. The first row of images depicts the SH fitting results, while the second row showcases the outcomes of interpolation. It reveals that the SH fitting provides more informative radiance prediction results on the top line, while preserving the smoothness of distribution.**

**Table 2: MSE in the novel views between interpolation and SH fitting with different ratios between known rays and unknown rays.**

| Known/Unknown | 8:2 | 7:3 | 6:4 | 5:5 |
|---|---|---|---|---|
| Interpolation | 0.0652 | 0.0744 | 0.0835 | 0.0951 |
| SH Fitting | 0.0602 | 0.0686 | 0.0779 | 0.0881 |

Argoverse [5], and NuScenes [4]. These datasets are challenging for 3D reconstruction due to their object movement and lighting variations. From KITTI, we selected sequences with diverse driving conditions and reduced the data to simulate a 2.5 Hz capture frequency, using 25% of available frames. This subset is henceforth referred to as the 'sparse' dataset. In contrast, utilizing the entirety of the training set corresponds to a 'dense' data scenario. Similarly, for Argoverse, we halved the data to match its 5 Hz frequency. NuScenes provided additional complexity with its annotated 3D bounding boxes under various conditions. For each dataset, we reserved every tenth frame for testing and used the rest for training to assess our model's ability to handle sparse viewpoints effectively. The poses for image alignment were taken directly from the datasets' provided odometry and tracking data to maintain consistency with the actual scale of the scenes.

**Implementation Details** In general, we implement our ray augmentation strategy based on NeuS [48]. Previous work [32] has shown that the input encoding scheme is important for reconstructing an implicit representation. Thus, we compare and choose the

frequency encoding [47] for the position input, and sphere harmonics function encoding [13] for the direction. We find that by replacing the frequency encoding with SH function encoding for direct input, the rendered color can converge faster.

We train our model using the Adam optimizer with an initial learning rate of $1e-4$ and a learning rate warm-up strategy. We choose a batch size of 2048 and train our model for 150k steps, and each scene is trained on one NVIDIA A100 GPU. To extract mesh for ray augmentation, we run marching cubes for a spatial resolution of $64^3$ to get the vertices. The same algorithm is used for extracting mesh when we evaluate the reconstruction result.

**Evaluation Metrics.** To validate the photorealism of our synthesized views against the ground truth, we employ widely accepted metrics from the field of novel view synthesis. Specifically, we measure the Peak Signal-to-Noise Ratio (PSNR), which reflects the reconstruction accuracy in terms of image pixel intensities. The Structural Similarity Index Measure (SSIM) assesses the perceived quality of the synthesized images, accounting for texture and structural integrity. Additionally, the Learned Perceptual Image Patch Similarity (LPIPS) metric evaluates the similarity between synthesized and ground truth images based on deep features, providing an estimate of perceptual likeness.

For the evaluation of depth reconstruction quality, we follow established precedents and include the Mean Absolute Relative Error (ABSREL) and the Root Mean Squared Error (RMSE). These depth accuracy metrics provide a quantitative measure of the disparity between the estimated depth maps and the ground truth, with ABSREL focusing on the relative difference and RMSE giving the Euclidean distance error.

### 5.3 Experiment Results

**Quantitative Analysis.** We conduct quantitative comparisons across the KITTI [17], NuScenes [4], and Argoverse [5] datasets to evaluate the performance of our method against several established baselines, including NeRF [30], NSG [35], pixelNeRF [55], SUDS [46], MARS [49], Urban-NeRF [38], MipNeRF-360 [3], and Instant-NGP [31]. In the dense scenario, our method consistently outperforms all baselines, achieving the highest PSNR and SSIM scores, while maintaining the lowest LPIPS and ABSREL scores across almost all three datasets. This indicates our approach's superior capability in synthesizing photorealistic views and reconstructing depth with high fidelity to the actual driving scenes.

The advantage of our method becomes even more pronounced in the sparse setting, where the limitations of existing techniques are

**Table 3: Evaluation and comparison of various methods on the NuScenes dataset. The best and the second best results are shown in bold and underlined forms, respectively.**

| Methods | Dense | | | | Sparse | | | |
|---|---|---|---|---|---|---|---|---|
| | PSNR↑ | SSIM↑ | LPIPS↓ | ABSREL↓ | PSNR↑ | SSIM↑ | LPIPS↓ | ABSREL↓ |
| NeRF [30] | 25.32 | 0.798 | 0.505 | 0.190 | 19.88 | 0.762 | 0.535 | 0.212 |
| NSG [35] | 26.10 | 0.810 | 0.498 | 0.185 | 20.34 | 0.770 | 0.528 | 0.208 |
| pixelNeRF [55] | 26.58 | 0.820 | 0.490 | 0.182 | 20.76 | 0.776 | 0.521 | 0.204 |
| SUDS [46] | 27.01 | 0.828 | 0.483 | 0.178 | 21.07 | 0.782 | 0.516 | 0.202 |
| MARS [49] | 27.45 | 0.835 | 0.475 | 0.173 | 21.50 | 0.790 | 0.510 | 0.198 |
| Urban-NeRF [38] | 27.92 | 0.843 | 0.461 | 0.169 | 20.09 | 0.802 | 0.504 | 0.195 |
| MipNeRF-360 [3] | 28.30 | 0.850 | 0.463 | 0.165 | 18.04 | 0.788 | 0.508 | 0.197 |
| Instant-NGP [31] | 27.80 | 0.840 | 0.472 | 0.170 | 16.42 | 0.780 | 0.512 | 0.243 |
| Ours | **29.50** | **0.862** | **0.430** | **0.160** | **26.92** | **0.868** | **0.450** | **0.182** |

**Table 4: Evaluation and comparison of various methods on the Argoverse dataset. The best and the second best results are shown in bold and underlined forms, respectively.**

| Methods | Dense | | | | Sparse | | | |
|---|---|---|---|---|---|---|---|---|
| | PSNR↑ | SSIM↑ | LPIPS↓ | ABSREL↓ | PSNR↑ | SSIM↑ | LPIPS↓ | ABSREL↓ |
| NeRF [30] | 26.54 | 0.812 | 0.491 | 0.185 | 21.44 | 0.785 | 0.521 | 0.205 |
| NSG [35] | 27.12 | 0.823 | 0.480 | 0.175 | 22.02 | 0.795 | 0.517 | 0.203 |
| pixelNeRF [55] | 27.64 | 0.831 | 0.477 | 0.175 | 22.63 | 0.805 | 0.502 | 0.196 |
| SUDS [46] | 27.88 | 0.835 | 0.462 | 0.165 | 23.01 | 0.814 | 0.495 | 0.190 |
| MARS [49] | 27.92 | 0.843 | 0.469 | 0.162 | 23.50 | 0.826 | 0.485 | 0.185 |
| Urban-NeRF [38] | 28.00 | 0.845 | 0.445 | 0.155 | 23.86 | 0.825 | 0.482 | 0.180 |
| MipNeRF-360 [3] | 29.35 | 0.855 | 0.446 | 0.120 | 25.81 | 0.829 | 0.468 | 0.139 |
| Instant-NGP [31] | 28.07 | 0.847 | 0.450 | 0.493 | 22.18 | 0.816 | 0.494 | 0.593 |
| Ours | **30.25** | **0.865** | **0.430** | **0.110** | **29.50** | **0.874** | **0.450** | **0.130** |

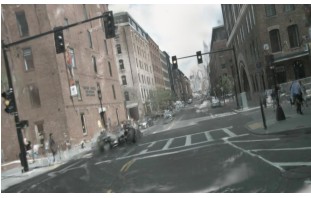
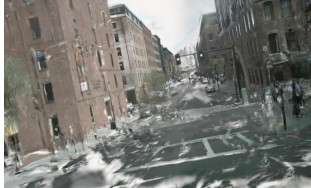
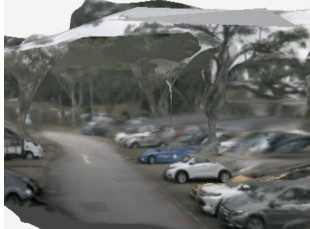
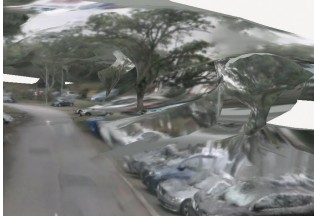

HarmonicNeRF                               MipNeRF-360

**Figure 4: Visual comparison of scene reconstruction on the NuScenes dataset, contrasting HarmonicNeRF with MipNeRF-360 [3].**

more apparent due to reduced input data. Specifically, on the KITTI dataset, our method surpasses the second-best performing method, MipNeRF-360, by a significant margin in terms of PSNR (19.04 vs. 16.93) and SSIM (0.672 vs. 0.589), highlighting its effectiveness in dealing with sparse data. Similar trends are observed on the NuScenes and Argoverse datasets, where our method demonstrates exceptional performance, particularly in terms of PSNR and SSIM, further underscoring its robustness and adaptability to various driving scenarios. These results not only underscore the efficacy of HarmonicNeRF in enhancing the quality of synthesized views and the accuracy of depth reconstructions under both data-rich

and data-sparse conditions but also set a new benchmark for future research in 3D scene reconstruction from sparse views.

**Qualitative Comparisons.** In addition to quantitative benchmarks, qualitative assessments on the KITTI dataset showcase the visual enhancements achieved with HarmonicNeRF. The comparison in Figure 5 reveals that, while Instant-NGP and MipNeRF-360 grapple with artifacts and blurring, particularly in motion-affected areas, HarmonicNeRF produces reconstructions with remarkable clarity and detail. Our method demonstrates superior texture fidelity and depth accuracy, capturing the nuances of urban driving scene complexity with greater photorealism.

On the NuScenes dataset, the superiority of HarmonicNeRF is further evidenced through visualizations presented in Figure 4. HarmonicNeRF excels in reconstructing densely populated urban areas, preserving distinct boundaries between objects, as shown on the left side of each comparison. In contrast, the right side displays results from MipNeRF-360, which, while competent, blends complex structures together—noticeably, trees meld with vehicles, and the finer architectural features are lost to blurring.

**Ablation Study.** We dissect the components of HarmonicNeRF to understand their individual contributions, especially in sparse data contexts where the intricacies of each module are crucial. The ablation study results in Figure 6 highlight the impact of the key features under both dense and sparse conditions.

**Without Spherical Harmonics.** Replacing spherical harmonics with linear interpolation caused a noticeable performance drop. In sparse scenarios, this led to a substantial decrease in PSNR (e.g., by 2.8) and SSIM (e.g., by 0.03), validating the efficacy of spherical harmonics in handling limited data.

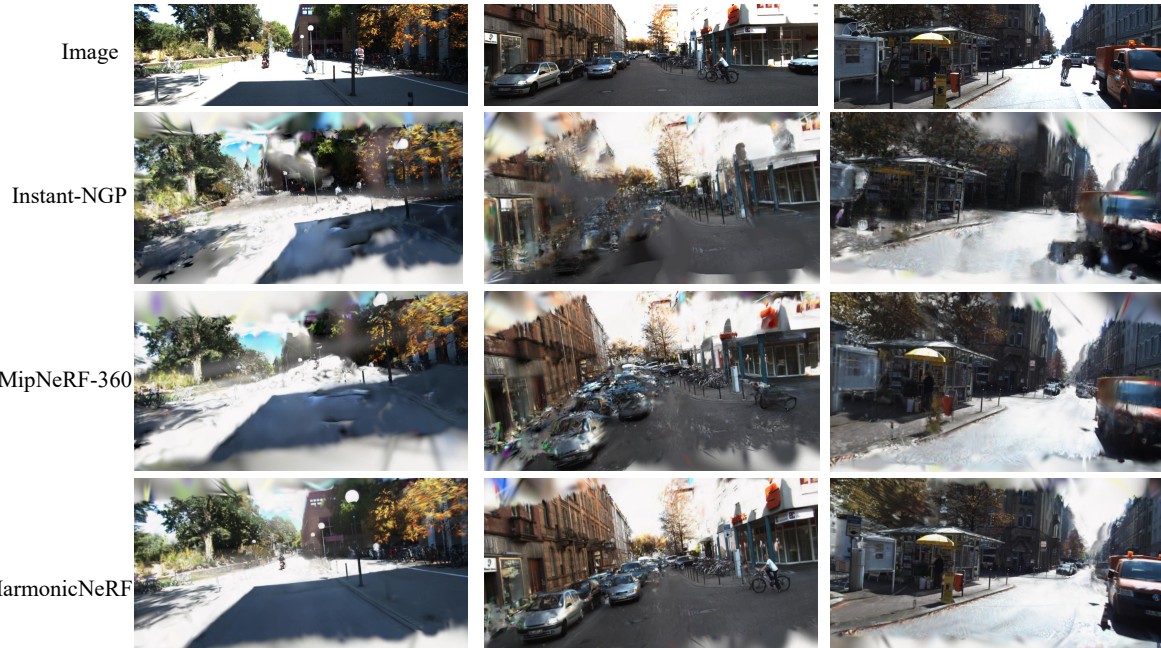

**Figure 5: Qualitative comparison of 3D scene reconstructions from the KITTI dataset using different neural radiance field methods. The top row presents original images from diverse urban settings with varying levels of detail and complexity. HarmonicNeRF, which consistently provides the clearest and most accurate depictions, with crisp textures and fine details, effectively handling challenging lighting and occlusions, and showcasing a marked improvement in both the fidelity and photorealism of the reconstructed scenes.**

**Without Visibility Checking.** Without this component, our model was unable to effectively handle occlusions, with a significant increase in LPIPS (e.g., by 0.05) and a more pronounced decrease in SSIM in sparse data (e.g., by 0.04), confirming the crucial role of visibility checking in sparse view synthesis.

**Alternative Ray Casting Methods.** Substituting our surface-guided ray casting with random ray casting resulted in less accurate reconstructions, particularly under sparse conditions, where PSNR dropped by 3.0 and LPIPS increased by 0.04.

**Without Depth Warping.** Removing depth warping adversely affected depth estimation accuracy, especially in sparse setups, with ABSREL rising by 0.1, which points to the depth warping's substantial role in enhancing depth precision.

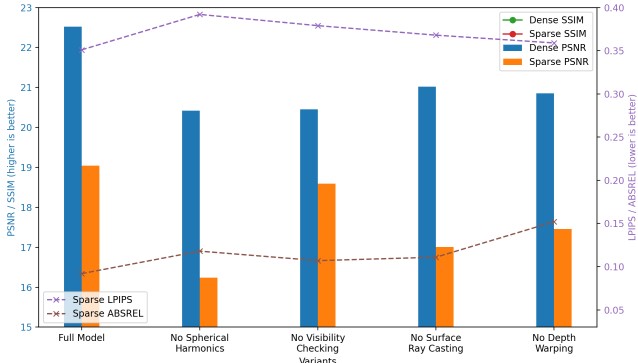

**Figure 6: Ablation study results on the KITTI dataset, illustrating the impact of each component under dense and sparse conditions.**

## 6 Conclusion

In this paper, we presented HarmonicNeRF addressing the significant challenge of sparse view reconstruction in the context of autonomous driving. Our approach innovatively enhances NeRF by integrating a geometry-guided ray augmentation strategy. This strategy not only employs a visibility check to filter out non-contributory rays using coarse geometry but also leverages spherical harmonics to model the natural distribution of radiance. This dual approach ensures physically plausible augmentations and superior surface reconstruction accuracy by effectively utilizing the limited observational data typical of unbounded outdoor scenes encountered in autonomous driving.

Our comprehensive experiments, spanning the KITTI, Argoverse, and NuScenes datasets, underscore HarmonicNeRF's ability to outperform existing methods significantly. By synthesizing novel depth views and reconstructing scenes with unprecedented precision, our method demonstrates its potential to set new standards for 3D scene reconstruction in autonomous driving applications.

However, limitations exist in our current framework. For instance, regions occluded in all observational views, such as the backside of objects, remain a challenge due to our visibility check's inability to augment such areas. Future endeavors could explore the integration of global scene priors or learning-based methods to infer these occluded regions.

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
