# OpenReview forum: "HarmonicNeRF: Geometry-Informed Synthetic View Augmentation for 3D Scene Reconstruction in Driving Scenarios"
_acmmm.org/ACMMM/2024/Conference — MM2024 Poster_

### Official Review · Reviewer_49nK · 2024-05-16

**Rating:** 3
**Confidence:** 2

**Summary:**

The authors propose HarmonicNeRF, a novel approach for self-supervised monocular scene reconstruction in driving scenarios. HarmonicNeRF enhances surface reconstruction accuracy by augmenting input space with geometry-informed synthetic views, using spherical harmonics to generate novel radiance values based on limited real-world color observations. This augmentation method incorporates proxy geometry to employ a visibility check and generate radiance pseudo-labels. The experiments implemented on the KITTI, Argoverse, and NuScenes datasets show that HarmonicNeRF outperforms baselines.

**Strengths:**

1. The authors clearly introduce the difficulties in designing each module and their solutions.

2. The authors provide some illustrations and visualization results to illustrate their design and performance.

**Limitations:**

1. Confusing contribution for dynamic scene reconstruction. The authors introduce difficulties like sparse views and dynamic scenes and show the experimental results in driving scenarios. However, the authors don't explicitly design modules tailored for dynamic conditions. Consequently, it's hard to determine whether HarmonicNeRF effectively mitigates the dynamic conditions or just takes effect in sparse-view conditions.

2. The motivation for this work in the context of driving scenarios is not sufficiently articulated. The related algorithms discussed in the paper, particularly in the introduction and related work sections, should be more closely aligned with driving scenarios. While the classical challenges such as dynamic conditions and constrained capturing trajectories in driving scenarios are well-known, and the ability of NeRF-related methods to address dynamic conditions in generic scenarios is acknowledged, it would be beneficial for the authors to provide deeper insights into the specific limitations of existing methods in driving scenarios.

3. The persuasiveness of the proposed method would be significantly enhanced by comparing it to more competitive approaches within the driving scenario context. The authors should consider comparing HarmonicNeRF to related works such as LocalRF, READ, and S-NeRF, which are mentioned in the paper. If there are any difficulties in making these comparisons, the authors should clarify them. Additionally, it is somewhat perplexing that algorithms specifically designed for scenarios like Urban-NeRF and SUDS do not significantly outperform NeRF and some variants.

4. The proposed method relies on rough scene geometry derived from a hand-crafted training strategy, which may limit its broad applicability. It would be beneficial to provide additional experiments demonstrating the robustness of the method. For instance, the authors could illustrate how the proposed method performs with varying qualities of the signed-distance function (i.e., initial scene geometry).

(BTW, the best and the second best results of ABSREL in Table 2 are mislabeled.)

- In summary, my primary concerns lie with the technical motivation and the benchmark design. I would currently rate this work "borderline reject". However, I am open to increasing my rating if the authors can adequately address these concerns. :)

**Suitability:**

2

---

### Official Review · Reviewer_5sSB · 2024-05-24

**Rating:** 4
**Confidence:** 3

**Summary:**

This paper introduces a new technology called HarmonicNeRF, which is a Geometry-Informed Synthetic View Augmentation for 3D Scene Reconstruction in driving scenarios. This technique is based on NeRF and enhances it by using spherical harmonics to generate new radiance values, combined with color observations from a limited number of real-world views, to improve the accuracy of surface reconstruction. Moreover, the method also uses proxy geometry to effectively manage occlusion issues, generating radiance pseudo-labels that address the limitations of traditional image-warping techniques, which often fail in the sparse data conditions typical of autonomous driving environments.

The authors conducted extensive experiments on autonomous driving-related datasets such as KITTI, Argoverse, and NuScenes, proving that their method significantly outperforms existing methods in synthesizing new depth views and reconstructing scenes, setting a new benchmark for 3D scene reconstruction in the field of autonomous driving.

**Strengths:**

1. The paper introduces a novel approach to enhancing NeRF by integrating geometry-informed synthetic views. This creative solution addresses the challenge of sparse viewpoints in 3D scene reconstruction for autonomous driving scenarios. The use of spherical harmonics for radiance distribution modeling and proxy geometry for occlusion management represents a new direction in NeRF-based scene reconstruction.
2. The technical approach of HarmonicNeRF is sound, including detailed procedures for surface ray casting, visibility checking, and radiance map estimation. The integration of these components into a cohesive framework demonstrates a deep understanding of the underlying technology and the challenges it aims to address.
3.  The paper provides a comprehensive evaluation using established datasets like KITTI, Argoverse, and NuScenes. It employs standard metrics such as PSNR, SSIM, LPIPS, and ABSREL for quantitative analysis, which are appropriate for assessing image quality and depth accuracy. Qualitative comparisons and ablation studies further validate the method's effectiveness.
4. The paper is well-structured and clearly articulates the problem, proposed solution, and results. Visual aids like figures and tables enhance the clarity of the presentation. The methodology is described step-by-step, making it easy for readers to follow.

**Limitations:**

1.3D-GS has been very popular in the field of 3D reconstruction recently, and it would be  better if more comparisons with 3D-GS based methods could be demonstrated.
2.Hope you can specifically write about the relationship between the paper and multimedia as well as multimodal processing, and the reasons for submitting to this conference.

**Suitability:**

2

---

### Official Review · Reviewer_r3W1 · 2024-05-25

**Rating:** 4
**Confidence:** 3

**Summary:**

The paper proposed a method called HarmonicNeRF to address the challenges during scene reconstruction posed by sparse viewpoints in autonomous driving scenarios. It first reconstruct a coarse mesh and and cast random rays from the mesh vertices to generate rays of unseen viewpoints, and visibility check is applied to ensure the generated rays are valid. Then, the radiance value of the rays from unseen viewpoints are calculated to bring more supervision to the scene. To maintain view-dependent affect, the radiance values of unseen rays are learned by fitting spherical harmonics on the known rays. Additional depth supervision of the unseen views are provided through depth warping from the known views. Through the above process, the paper demonstrate state-of-the-art reconstruction performance KITTI, Argoverse and NuScenes.

**Strengths:**

- The paper solves the problem of scene reconstruction from sparse viewpoints in autonomous driving scenario by providing additional supervision signal for unseen viewpoints.
- The radiance value of the unseens rays are learned by fitting spherical harmonics on the known rays in the input images.
- Extensive experiments on the KITTI, Argoverse, and NuScenes datasets demonstrate that HarmonicNeRF outperforms existing methods.
- The paper is well-written and easy to follow.

**Limitations:**

- Missing an overall loss function.
- For ablation study, it would be better to present a table to show the comparision of  numerical results.
- In section 5.2, the paper mentions that for depth reconstruction quality, it will report both ABSREL and RMSE, but only ABSREL is reported.
- The paper doesn't mention where the depth maps used for depth warping come from, are they obtained from the trained NeRF or from a depth sensor?

**Suitability:**

2

---

### Official Review · Reviewer_TiJk · 2024-06-01

**Rating:** 5
**Confidence:** 3

**Summary:**

This paper introduces HarmonicNeRF, a novel approach to 3D scene reconstruction tailored for autonomous driving scenarios. By integrating spherical harmonics and proxy geometry, this method generates synthetic views that overcome the typical limitations of sparse and path-constrained data in autonomous driving data collection. HarmonicNeRF demonstrates superior reconstruction accuracy and robustness across multiple benchmark datasets, enhancing reconstruction quality and paving the way for technological advancements in similar application fields.

**Strengths:**

1. The logic in the abstract and introduction sections is clear and well-articulated.
2. The figures in the paper are accompanied by detailed explanations.
3. The paper validates the proposed method through extensive experiments on multiple standard autonomous driving datasets, including KITTI, Argoverse, and NuScenes. These experiments not only demonstrate the effectiveness of the method but also compare its performance with current state-of-the-art techniques.
4. The paper provides a detailed description of each component of the method, including the application of spherical harmonics and the use of proxy geometry. This thorough explanation aids readers in comprehensively understanding the working principles and implementation details of the method.
5. The paper explicitly outlines its technical contributions and potential application areas, guiding future research directions, particularly in autonomous driving and other applications requiring high-precision 3D reconstruction.

**Limitations:**

1. The paper mentions that the proposed method is a plug-and-play approach. I believe it is necessary to demonstrate this with relevant experiments, including additional ablation studies with baselines such as MipNeRF and PixelNeRF.
2. The paper introduces complex techniques involving spherical harmonics and proxy geometry, which, while effective, may be computationally intensive. The paper could discuss the computational costs in more detail, including training time and inference speed, especially considering real-time applications. Additional experiments might be needed to supplement this discussion.
3. There seems to be an issue with Equation 7 in Section 4.4: $ p_j = K_{\text{unseen}}^{-1} \left[Z_j^{-1}\right]^T \left(K_{\text{ref}}^{-1} Z_i p_i\right)$ is missing an inverse for $ Z_j $. It is advisable to double-check this formula.

**Suitability:**

2

---

### Meta-Review · Area_Chair_PEVv · 2024-07-01

**Recommendation:** Accept (Poster)
**Confidence:** 4

**Metareview:**

Overall, authors have clearly stated their motivation and proposal, allowing the reviewers to understand the paper better. Furthermore, the authors have explained their methodology and presented their results well. Moreover, according to reviewers, the authors have responded to initial reviews with adequate response to justify their proposal.